# MixFilter: Pre-train Aware Structured Dropout for Domain Generalization

## Abstract

Model ensembling is a widely adopted technique for improving the robustness of convolutional neural network (CNN) classifiers against distribution shifts. This method involves either averaging the predictions of multiple models or combining their weights. However, it comes with considerable computational overhead, as it requires training multiple networks. Recently, fine-tuning with very high dropout rates at the penultimate layer has been shown to mimic many benefits of ensembling without requiring multiple training runs. However, a performance gap persists, likely due to the limited regularization applied solely at the final layer of CNNs. In this paper, we present MixFilter, a novel dropout strategy that is designed for fine-tuning convolutional neural networks that leverage rich pre-trained representations for domain generalization. MixFilter enhances functional diversity across subnetworks by stochastically mixing convolutional filters from all the layers of fine-tuned and pre-trained models. Our experimental results indicate that on five domain generalization benchmarks—PACS, VLCS, OfficeHome, TerraIncognita, and DomainNet—MixFilter achieves out-of-domain accuracy comparable to ensemble-based approaches while avoiding additional inference or training overhead. Anonymized source code is available at `https://anonymous.4open.science/r/MixFilter-6EEE`.

## 1 Introduction

Convolutional neural networks (CNNs) LeCun et al. (1989) have revolutionized image classification, leading to numerous breakthroughs Krizhevsky et al. (2017); Szegedy et al. (2015); He et al. (2016). However, their performance can degrade significantly when there is a substantial distribution shift between the training and test data (Koh et al., 2021). To address this issue, domain generalization (DG) aims to develop methods that train models on a limited number of source domains and evaluate them on distinct, unseen domains (Zhou et al., 2022).

Among state-of-the-art DG techniques, ensembling (Lakshminarayanan et al., 2017) and weight averaging (Wortsman et al., 2022a) stand out in terms of their performance on diverse DG benchmarks under fair evaluation protocols in realistic settings (Gulrajani & Lopez-Paz, 2020; Rame et al., 2022; Arpit et al., 2022). Ensembling combines predictions from multiple models, whereas weight averaging merges the parameters of several models into a single one before making predictions. These approaches can significantly outperform other DG methods by leveraging the diversity among multiple models. However, achieving such outstanding performance necessitates numerous training sessions with varied hyperparameters and initializations. This requirement can become prohibitively expensive, especially when dealing with large-scale datasets.

An effective yet straightforward method to enhance the generalization of a model is deactivating some neurons during training, as demonstrated by Dropout (Srivastava et al., 2014). Dropout acts as an implicit ensemble by only updating sparse subnetworks during training. Despite its efficacy, Dropout has two significant limitations. First, randomly dropping features works well for fully connected layers but is less effective for convolutional layers due to the spatial correlations between features. This spatial correlation still allows substantial input information to propagate to the next layer, diminishing the regularization effect. Second, a powerful pre-trained model is crucial for the success of DG methods (Wiles et al., 2021; Koh et al.,

2021), and Dropout does not address the integration of such models. To overcome these issues, structured variants of Dropout, such as SpatialDropout (Tompson et al., 2015) and DropBlock (Ghiasi et al., 2018), have been introduced to better regularize convolutional networks by focusing on spatially correlated features. Additionally, Mixout (Lee et al., 2019) proposes a stochastic blending of MLP parameters from fine-tuned and pre-trained models, enhancing the use of pre-trained networks. However, no approach in DG effectively addresses both drawbacks simultaneously for CNNs.

In this paper, we present MixFilter, a novel Dropout strategy that enhances Standard Dropout in CNNs for out-of-domain settings by adhering to three main principles. ① MixFilter incorporates pre-trained knowledge into the dropout process by stochastically blending filters from both fine-tuned and pre-trained models. ② The choice of working on filters rather than relying on random activation, as in standard dropout, enables effective information regularization within the CNN structure. ③ Blending filters in the weight space, rather than the activation space, makes MixFilter computationally more efficient. MixFilter's effectiveness stems from creating an implicit ensemble through numerous subnetworks with extensive weight sharing. Unlike Standard Dropout, these subnetworks are not sparse and heavily rely on pre-trained weights, facilitating knowledge transfer to the target task without overfitting. Furthermore, unlike traditional ensemble methods that require training multiple models, MixFilter enhances network robustness within a single training session.

**Our main contributions can be summarized as follows.**

1. We identify three key principles that enable Standard Dropout on CNNs to achieve ensemble-level performance in the DG setting with a single training run. Based on these principles, we introduce MixFilter, a pre-trained-aware Dropout variant that enhances CNN robustness against distribution shifts.

2. Through numerous ablation studies, our design choices are validated and underscore the advantages of MixFilter compared to other dropout variants.

3. Empirical results indicate that, on average, MixFilter achieves out-of-domain accuracy comparable to ensemble and weight averaging methods on five DG benchmarks from DomainBed, all without incurring any additional inference or training overhead.

## 2 Related Work

### 2.1 Dropout

Dropout (Srivastava et al., 2014) is a regularization method where, during training, a random subset of neuron activation is set to zero, effectively deactivating them. This introduces noise into the neural network, preventing overfitting by ensuring each training sample is processed by a different sub-network. All neurons are active during inference, but their outputs are scaled to match the training conditions. This process can be seen as training an implicit ensemble of many sub-networks, improving the model's generalization. Dropout has inspired various techniques to inject noise into neural networks (Ferianc et al., 2024) by randomly deactivating entire layers (Huang et al., 2016; Larsson et al., 2016), channels (Pan et al., 2020; Tompson et al., 2015), neuron connections (Wan et al., 2013), or contiguous regions of a feature map (Ghiasi et al., 2018), all aimed at preventing overfitting by introducing randomness. These methods typically apply Dropout during training from scratch. However, using Dropout directly in pre-trained models can disrupt learned representations, making the pre-trained weights less effective (Zhang & Bottou, 2024). Mixout (Lee et al., 2019) addresses this for MLPs by stochastically mixing pre-trained and fine-tuned parameters. For CNNs, where structured noise is more effective, our method, MixFilter, extends this idea by mixing entire filters from pre-trained and fine-tuned models. This allows for retaining pre-trained features while adapting to new tasks without losing the benefits of pre-training.

### 2.2 Domain Generalization

State-of-the-art approaches to DG can be broadly categorized into three main strategies: (1) *regularization of feature and predictor*, (2) *data augmentation*, and (3) *leveraging pre-trained models*.

**Regularization of feature and predictor.** DANN (Ganin et al., 2016) utilizes adversarial networks to ensure features from different domains are statistically indistinguishable. This foundational work has led to various other methods that apply regularization to the feature space such as minimizing the maximum mean discrepancy (Li et al., 2018a), invariance of the conditional distribution (Li et al., 2018b; Albuquerque et al., 2019), and invariance of the covariance matrix of the feature distribution (Sun & Saenko, 2016). Instead of regularizing the feature space, techniques like IRM (Arjovsky et al., 2019) enforce the same optimal classifier across different domains. Fish (Shi et al., 2021) and IGA (Koyama & Yamaguchi, 2020), introduce gradient alignment constraints to ensure consistency across training environments. GroupDRO (Sagawa et al., 2019) addresses DG by focusing on minimizing the worst-case training loss and prioritizing more challenging domain samples during training. Meta-learning approaches, as explored in Bui et al. (2021), adapt model parameters to new domains during training. Despite these innovations, achieving true invariance is often challenging and can be overly restrictive (Zhao et al., 2019). The effectiveness of these regularization techniques is mixed, as evidenced by the strong performance of Empirical Risk Minimization (ERM) (Gulrajani & Lopez-Paz, 2020), suggesting that some regularizations may be either too strong to optimize reliably or too weak to meet their objectives (Zhang et al., 2022).

**Data augmentation.** Data augmentation is another powerful strategy to enhance DG by expanding the diversity of the training dataset. Techniques such as RandAugment (Cubuk et al., 2020), TrivialAugment (Müller & Hutter, 2021), AugMix (Hendrycks et al., 2019), and MixUp (Zhang et al., 2017) create robust models by introducing variability in the training data. Beyond heuristic augmentations, some methods leverage domain meta-data to learn challenging and diverse transformations (Zhou et al., 2020; Yan et al., 2020; Aminbeidokhti et al., 2024) or synthesize novel domains through style mixing (Zhou et al., 2021) or generative models (Goel et al., 2020).

**Leveraging pre-trained models.** Several recent approaches aim to leverage generalizable features from a model pre-trained on large-scale data. Adapting these models without forgetting their broad and versatile representations is key to achieving generalization in downstream tasks. Model soups (Wortsman et al., 2022a) and DiWA (Rame et al., 2022) use weight averaging to combine the properties of diverse fine-tuned networks. (Cha et al., 2021; Arpit et al., 2022; Wortsman et al., 2022b) maintain a running average of model parameters during training, effectively creating an ensemble of the initial and fine-tuned models. To prevent feature distortion, Kumar et al. (2022) propose pre-training a linear probe before fine-tuning the model backbone. MIRO (Cha et al., 2022) maximizes the mutual information in feature space between the fine-tuned and pre-trained networks.

Recent research (Zhang & Bottou, 2024) shows that applying Standard Dropout within a nonlinear deep network introduces complex noise patterns, which can hinder the development of internal representations and stall the optimization process. To address this, they employed a very high Dropout rate at the penultimate layer during the fine-tuning of large pre-trained models, effectively narrowing the performance gap between single-model training and ensemble methods. However, our experiments indicate that using MixFilter not only enables Dropout in a nonlinear deep network but also significantly boosts generalization, often matching or surpassing the performance of ensemble-based techniques.

## 3 MixFilter: Pre-train Aware Structured Dropout

We aim to adapt the Dropout mechanism to enhance its suitability for DG tasks. To achieve this, we focus on three key principles: ① *leveraging pre-trained knowledge through mixing fine-tuned and pre-trained activations*, ② *effective information regularization through structured masking*, and ③ *computational efficiency by switching to weight space*. This section reviews the Standard Dropout mechanism and then elaborates on the design choices that align with each of these principles.

**Standard Dropout.** Consider a convolutional layer in a neural network. Let $X \in \mathbb{R}^{H \times W \times C_{in}}$ and $Y \in \mathbb{R}^{H' \times W' \times C_{out}}$ represent the input and the output (activations) tensors where $H, W, H', W'$ show the spatial dimensions and $C_{in}, C_{out}$ demonstrate the number of channels. Let $W \in \mathbb{R}^{k \times k \times C_{in} \times C_{out}}$ represent convolutional filter weights with kernel size of $k$. The activations $Y$ are computed by convolving the weight matrix $W$ over the input $X$, followed by applying a non-linear activation function $a$, such as ReLU. During

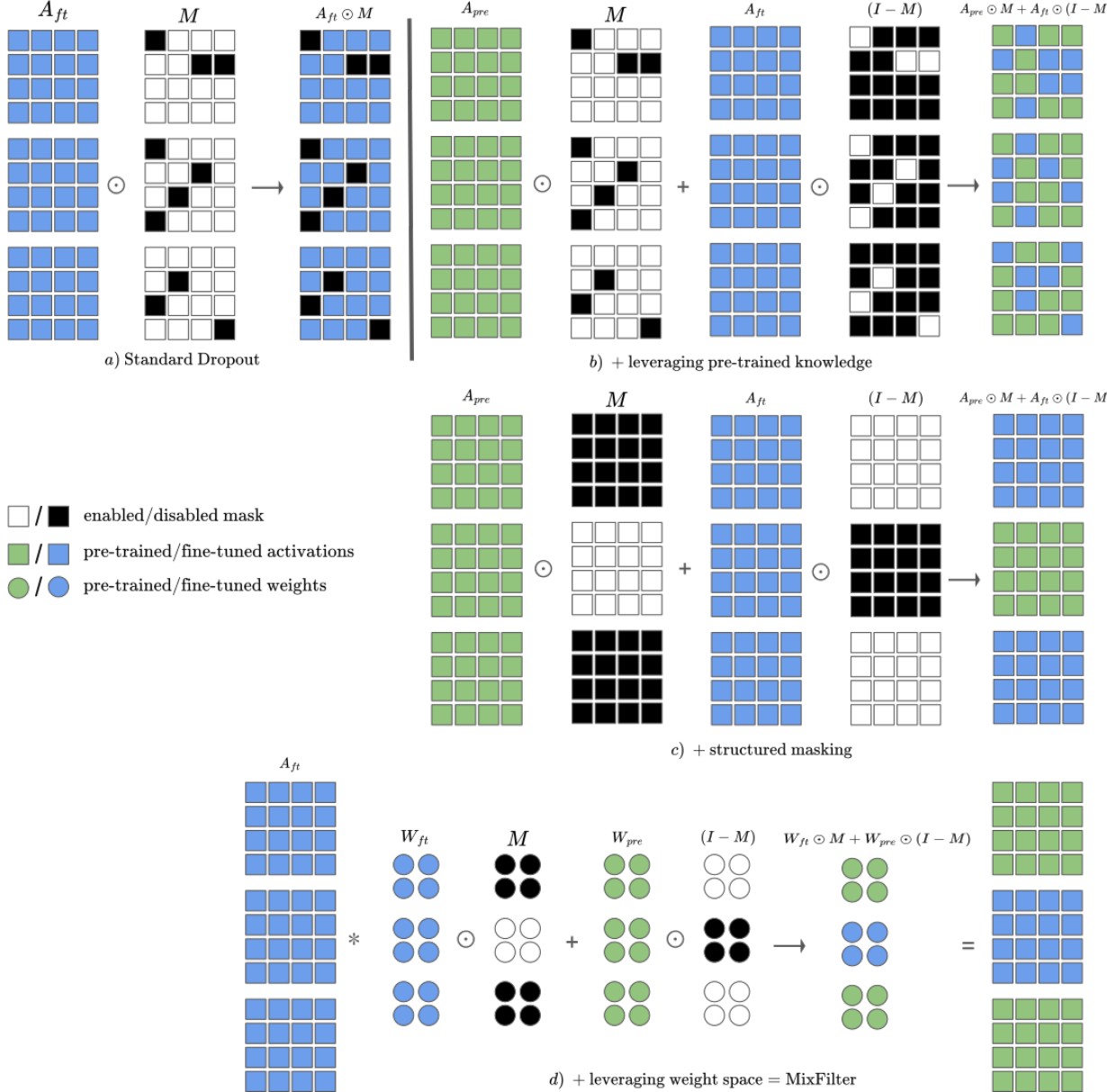

Figure 1: Components of MixFilter. In this diagram, pre-trained and fine-tuned information is represented in green and blue, respectively. Squares depict activations, while circles represent convolutional filters. (a) Standard Dropout is applied to the activation space. (b) Include pre-trained knowledge with a mix of pre-trained and fine-tuned activations. (c) Structured masking to enhance the regularization of information propagation in CNNs. (d) Transition to the weight space to ensure computational efficiency.

training, the Standard Dropout technique randomly retains each element of the layer's output with probability $p$. Elements not retained are set to zero with a probability of $1 - p$. In practice, an inverted version of Dropout is used to maintain the same expected output scale, where the activations are scaled by $\frac{1}{1-p}$ during training. This operation can be expressed mathematically as:

$$Y = \frac{a(W * X) \odot M}{1 - p},\tag{1}$$

where $\odot$ denotes element-wise product, $*$ denotes convolution operation, and $M$ is a binary mask matrix of the same size as $Y$ with each element drawn independently from the Bernoulli distribution with a mean equal to $p$. Normally we tune $p$ using the validation set. In Figure 1a, you can find the visualization of Standard Dropout on activations of a convolutional layer. In the figure, we drop the scaling factors for simplifications.

**Leveraging pre-trained knowledge through mixing fine-tuned and pre-trained activations.** Pre-training on a large dataset is widely recognized as a crucial technique to mitigate performance degradation caused by distribution shifts across different datasets (Wiles et al., 2021). However, the Standard Dropout mechanism does not inherently utilize the valuable information embedded in pre-trained models. To address this limitation, a straightforward approach is to replace the dropped activations with corresponding activations derived from a pre-trained model, instead of setting them to zero (Lee et al., 2019). This modification enables us to infuse the network with pre-trained knowledge during Dropout. The Standard Dropout operation can be redefined to incorporate this pre-trained information as follows:

$$Y = \frac{\textcolor{red}{a(W_{pre} * X) \odot (I - M)} + a(W_{ft} * X) \odot M - \textcolor{blue}{a(W_{pre} * X) \cdot p}}{1 - p},\tag{2}$$

where $I$ is the identity vector and $W_{ft}$ and $W_{pre}$ denotes the fine-tuned and pre-trained parameters respectively. In the equation 2 there are two main differences compared to Standard Dropout in equation 1. The red part handles the pre-trained knowledge infusion and the blue part ensures that the expectation of the output during training and inference stays the same. This process is illustrated in Figure 1b. As before, the figure is simplified by dropping scaling factors.

**Effective information regularization through structured masking.** While randomly dropping features, as in Standard Dropout, can be effective for fully connected layers, its efficacy diminishes when applied to convolutional layers due to the strong spatial correlations among features. These correlations allow significant information about the input to propagate through the network even when some features are masked, which can lead to overfitting. This highlights the need for a more structured and spatially-aware approach for masking to effectively regularize convolutional networks. To address this issue in convolutional layers, two effective methods are SpatialDropout (Tompson et al., 2015) and DropBlock (Ghiasi et al., 2018). SpatialDropout randomly drops entire channels from a feature map, while DropBlock removes a contiguous region of the feature map. As we will explain below, we opt for SpatialDropout because it is computationally more efficient to integrate with pre-trained models. Figure 1c illustrates the impact of structured masking on a sample feature map, showcasing how this strategy disrupts the spatial information to regularize the network.

**Computational efficiency by switching to weight space.** Ensemble and weight averaging methods have shown superior results in DG compared to regularization and data augmentation-based techniques (Rame et al., 2022; Arpit et al., 2022). However, these methods come with significant training overheads. Creating an effective ensemble requires training multiple models initialized with different hyper-parameters and weights, leading to considerable computational demands. Instead, we aim to use a method that does not add computation to training and inference. Following the standard formulation, using pre-trained knowledge in Dropout requires forward passes through both the pre-trained and fine-tuned models to obtain activations, which doubles the training computation. We overcome this by applying Dropout to the weight connections instead of the activations. This modification allows us to implement Dropout before the forward pass, eliminating the need to run the pre-trained and fine-tuned models separately. By integrating Dropout directly into the weight connections, we streamline the process, enhancing efficiency. This approach can be

expressed mathematically in terms of the weights as follows:

$$\bar{W}_{ft} = \frac{W_{pre} \odot (I - M) + W_{ft} \odot M - W_{pre} \cdot p}{1 - p} \tag{3}$$

where $\bar{W}_{ft}$ is the new fine-tuned parameters after mixing the previous one with pre-trained weights. Mix-Filter can be viewed as a pre-trained-aware, structured variant of Dropout which stochastically mixes the convolutional filters from fine-tuned and pre-trained models. While it shares similarities with Mixout (Lee et al., 2019), which was initially developed for NLP tasks and primarily used in MLP layers, MixFilter is tailored specifically for CNNs. This approach is particularly designed to address the challenges of DG tasks in these models. Figure 1 shows the overview of MixFilter. In section 4.2, different design choices are explored for MixFilter to validate of our proposed method wrt other variants.

## 4 Experiments

**Datatsets.** Following DomainBed benchmark (Gulrajani & Lopez-Paz, 2020), we evaluate our method on five diverse datasets. PACS (Li et al., 2017) is a 7-way object classification task with 4 domains and 9,991 samples. VLCS (Fang et al., 2013) is a 5-way classification task with 4 domains and 10,729 samples. This dataset mostly contains real photos. The distribution shifts are subtle and simulate real-life scenarios well. OfficeHome (Venkateswara et al., 2017) is a 65-way classification task depicting everyday objects with 4 domains and a total of 15,588 samples. TerraIncognita (Beery et al., 2018) is a 10-way classification problem of animals in wildlife cameras, where the 4 domains are different locations. There are 24,788 samples. This represents a realistic use case where generalization is indeed critical. DomainNet (Peng et al., 2019) is a 345-way object classification task with 6 domains. With a total of 586,575 samples, DomainNet is larger than most of the other evaluated datasets in both samples and classes.

**Evaluation Protocol.** We report out-of-domain accuracies for each domain and their average using a leave-one-out cross-validation method. In this approach, each domain is sequentially used as the target (test) domain, while the remaining domains are utilized as source (training) domains. Our evaluation protocol adheres closely to the DomainBed framework for training and evaluation (Gulrajani & Lopez-Paz, 2020), with one notable modification: we employ the "IMAGENET1K_V2" variant from PyTorch (Paszke et al., 2019) as the pre-trained weights for ResNet50 backbone. This selection is motivated by the observation that robust pre-trained models can significantly outperform more sophisticated fine-tuning approaches (Zhang & Bottou, 2024; Wiles et al., 2021). We use Adam (Kingma & Ba, 2014) optimizer with a mini-batch containing all domains and 32 examples per domain. For the model hyperparameters, such as learning rate, dropout rate, and weight decay, we use the same configuration as proposed in Cha et al. (2021), as detailed in the Appendix A. We follow Cha et al. (2021) and train models for 15000 steps on DomainNet and 5000 steps for other datasets, corresponding to a variable number of epochs dependent on dataset size. Every experiment is repeated three times with different seeds. We leave 20% of source domain data for validation. We use training-domain validation for the model selection, in which, for each random seed, we choose the model, maximizing the accuracy of the validation set.

**Baselines.** In addition to Empirical Risk Minimization (ERM) (Vapnik, 1991) and Mixout (Lee et al., 2019), we include CORAL (Sun & Saenko, 2016) which is the best approach among domain invariance learning methods. We evaluate MixFilter against ensemble (ENS) (Lakshminarayanan et al., 2017) and weight averaging (DiWA) methods, which typically outperform ERM and other DG baselines but require many models to train (Cha et al., 2021; Rame et al., 2022). We also include model averaging (MA) variants of ERM and MixFilter which average checkpoints collected during a single fine-tuning process. Recently, Zhang & Bottou (2024) demonstrated that employing a significantly large dropout rate on the penultimate layer during fine-tuning markedly enhances the performance of deep neural networks, achieving results comparable to ensemble-based methods, particularly in out-of-distribution scenarios. In their study, they employ SGD optimizer as opposed to Adam with twice as many iterations per dataset compared to the DomainBed default configuration. We re-run their experiments under the same conditions and configurations, reporting the results as the "Large Dropout" method in Table 1. For each domain within a dataset, we first fine-tune

| Method | #Train | #Inf | PACS | VLCS | OfficeHome | TerraInc | DomainNet | **Avg.** |
|---|---|---|---|---|---|---|---|---|
| Mixout (Lee et al., 2019) | *1* | *1* | $86.36_{\pm0.72}$ | $79.52_{\pm0.49}$ | $71.02_{\pm0.58}$ | $49.28_{\pm1.86}$ | $47.40_{\pm0.21}$ | $66.72_{\pm0.77}$ |
| CORAL (Sun & Saenko, 2016) | *1* | *1* | $87.40_{\pm0.67}$ | $80.01_{\pm0.52}$ | $71.23_{\pm0.34}$ | $50.61_{\pm1.48}$ | $47.33_{\pm0.29}$ | $67.32_{\pm0.66}$ |
| Large Dropout (Zhang & Bottou, 2024) | *1* | *1* | $87.14_{\pm0.62}$ | $79.31_{\pm0.43}$ | $70.66_{\pm0.38}$ | $52.27_{\pm1.55}$ | $47.36_{\pm0.17}$ | $67.35_{\pm0.63}$ |
| ERM (Vapnik, 1991) | *1* | *1* | $87.66_{\pm0.71}$ | $79.64_{\pm0.43}$ | $70.46_{\pm0.70}$ | $52.62_{\pm2.33}$ | $48.48_{\pm0.48}$ | $67.77_{\pm0.93}$ |
| ERM (MA) (Arpit et al., 2022) | *1* | *1* | $88.25_{\pm0.38}$ | $79.86_{\pm0.32}$ | $71.97_{\pm0.16}$ | $54.59_{\pm0.85}$ | $49.05_{\pm0.06}$ | $68.47_{\pm0.35}$ |
| ENS (Lakshminarayanan et al., 2017) | *18* | *18* | 89.05 | 80.03 | 71.77 | 54.10 | 49.11 | 68.80 |
| DiWA (Rame et al., 2022) | *18* | *1* | 89.21 | 79.83 | 71.74 | 55.68 | 48.40 | 68.96 |
| **MixFilter** | *1* | *1* | $87.94_{\pm0.87}$ | $79.40_{\pm0.39}$ | $72.14_{\pm0.30}$ | $58.42_{\pm0.66}$ | $47.69_{\pm0.26}$ | $69.12_{\pm0.50}$ |
| **MixFilter (MA)** | *1* | *1* | $88.25_{\pm0.22}$ | $79.50_{\pm0.23}$ | $72.66_{\pm0.24}$ | $58.49_{\pm0.56}$ | $48.89_{\pm0.05}$ | $69.56_{\pm0.26}$ |

Table 1: Out-of-domain accuracy on five DG benchmarks from DomainBed. We show the number of models required for each method during training and inference. Average accuracy and standard error are reported from three trials. Results per domain are shown in Appendix C.

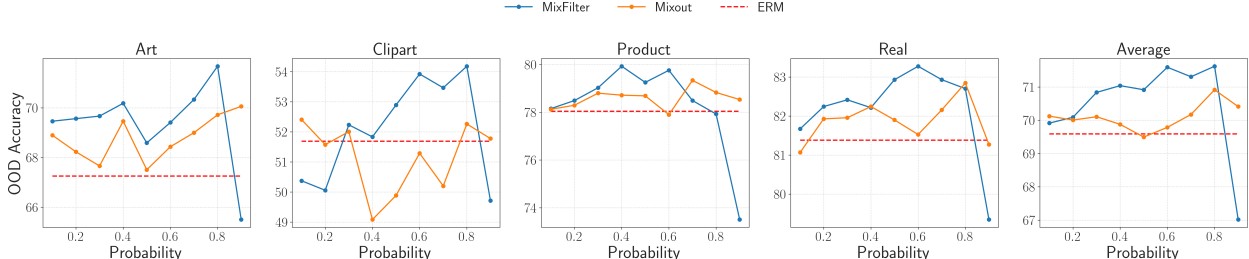

Figure 2: Comparison of MixFilter with Mixout on domains within the OfficeHome dataset, with ERM included as a reference. The figures illustrate out-of-domain accuracy across varying drop/mix probabilities for each method. We exclude SpatialDropout from the figure since it consistently underperformed relative to the ERM baseline and diverged when the drop probability exceeded 0.1. Mixout occasionally surpassed ERM performance but still did not match the performance of MixFilter across all tested domains.

16 models using a grid search to find the best hyperparameters for ERM, and then search for the optimal choice of parameter for each method. To evaluate ensemble-based methods, we use all of the 16 models in the hyperparameter search as the model pools. After hyperparameters tuning, we run each method on 2 more seeds and report the average performance.

## 4.1 Main Results

Table 1 presents a comparative analysis of MixFilter's performance against several other methods across five DG benchmarks for a classification task. Detailed results for each dataset and domain are available in the Appendix C. The table shows that MixFilter, along with its moving average variant, often matches the performance of ensemble-based methods. The columns labeled *#Train* and *#Inf* represent the number of models required by each method during training and inference, respectively. Unlike ENS and DiWA, which require 18 models, MixFilter needs only a single model for training, leading to a significant reduction in training time. Notably, on the TerraIncognita dataset, MixFilter achieves a significant performance lead over other baselines, all without the need for multiple training runs. TerraIncognita is recognized within the DomainBed benchmark suite for its challenging nature, featuring both high covariate and label shifts Chen et al. (2023). This superior performance suggests that MixFilter is adept at handling complex distribution shifts, showcasing its robustness and reliability in varied conditions.

## 4.2 Ablation Studies

In this section, experiments are performed on the OfficeHome dataset to gain a deeper understanding of the functionality and effectiveness of MixFilter. Unless stated otherwise, the experimental setups are consistent with those described in Section 4.

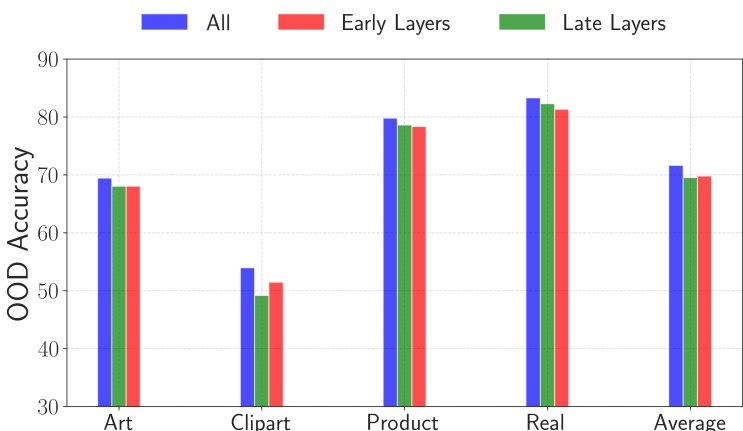

Figure 3: The optimal placement of MixFilter for the OfficeHome dataset. To assess the benefits across various levels of representation for DG tasks, we apply MixFilter to early, late, and all convolutional layers. Utilizing MixFilter across all convolutional layers not only yields superior performance compared to other configurations but also eliminates the need for additional hyper-parameters, reducing the overall complexity of the approach.

| MA | wdecay($W_{pre}$) | LP-FT | **ERM** | **MixFilter** |
|---|---|---|---|---|
| | | | 69.59 | 71.80 |
| ✓ | | | 71.61 | 72.30 |
| | ✓ | | 69.62 | 71.54 |
| | | ✓ | 69.68 | 71.06 |
| ✓ | ✓ | ✓ | 71.81 | 72.39 |

Table 2: Impact of various fine-tuning techniques on MixFilter and ERM. MA refers to model averaging and LP-FT is a two-stage fine-tuning where linear probing initializes the classifier head for the full fine-tuning stage. The last two columns show the out-of-domain accuracy of each method on the OfficeHome dataset across all domains. Unlike ERM, these techniques generally do not enhance MixFilter's performance, except for MA. The improvement in MA can likely be attributed to its more stable optimization process.

**Comparison with Dropout variants.** To evaluate the effectiveness of MixFilter, we compared it against SpatialDropout and Mixout applying to all convolutional layers within the network. Our findings are illustrated in Figure 2, which presents the performance across all domains of the OfficeHome dataset. These comparisons are conducted using the optimal hyper-parameters for ERM, as discussed in Section 4, and a varying probability for dropping or mixing, ranging from 0.1 to 0.9 in increments of 0.1. We omitted SpatialDropout from the figure because its performance consistently falls below the ERM baseline across all domains. Additionally, increasing the drop rate beyond 0.1 leads to training instability, frequently resulting in model divergence. Mixout, when applied with an appropriate mixing probability, surpasses the baseline performance. However, despite these enhancements, MixFilter consistently outperformed Mixout, delivering superior results across all domains.

Regarding the optimal mix rate, Mixout and MixFilter both benefit from higher mixing rates. However, Mixout, which lacks a structured approach in its masking process, does not gain as much advantage from increased mixing rates compared to MixFilter. Additionally, applying a mixing rate exceeding 0.8 tends to disable large portions of the network during the backward pass, leading to undertraining.

**Comparison with popular fine-tuning techniques.** Table 2 highlights the impact of MixFilter when combined with various fine-tuning techniques that use representations learned by pre-trained models to enhance the ability of fine-tuned models. It also compares this setup with the same for the ERM baseline. Notable fine-tuning techniques include (LP-FT) which involves tuning only the classifier head before fine-

| Sampling | A | C | P | R | **Avg.** |
|---|---|---|---|---|---|
| Bernoulli | 71.06 | 54.20 | 79.99 | 83.30 | 72.14 |
| Gaussian | 71.15 | 53.35 | 80.02 | 83.41 | 71.98 |

Table 3: Comparison of out-of-domain accuracy with Bernoulli and Gaussian distribution for sampling the mixing probability across domains from the OfficeHome dataset.

tuning the entire network (Kumar et al., 2022), Model Averaging (MA), which averages weights during training to improve robustness (Cha et al., 2021; Arpit et al., 2022), and weight decay directed toward the pre-trained model's parameters, referred to as wdecay ($W_{pre}$) in the table (Xuhong et al., 2018).

Previous studies (Kumar et al., 2022; Noci et al., 2024) show that using a smaller learning rate for the classifier head or tuning it separately can enhance OOD performance. This approach is beneficial because, during fine-tuning, it prevents the lower layers of the neural network from changing too much, thus preserving the quality of the pre-trained features. Contrary to this, we observe that LP-FT does not enhance the performance of MixFilter compared to ERM. We believe this is because MixFilter inherently mitigates feature distortion by effectively freezing a significant portion of the pre-trained features during each iteration, allowing for controlled and focused fine-tuning of the classifier head.

Penalizes deviations from the pre-trained weights using weight decay, is a widely-used technique to boost performance in transfer learning (Miceli Barone et al., 2017; Kirkpatrick et al., 2017). However, following the same virtue as in (Lee et al., 2019), MixFilter can be interpreted as applying an adaptive $L_2$-penalty towards the pre-trained weights. Table 2 illustrates that while applying wdecay ($W_{pre}$) improves the performance of ERM, it still falls short compared to the effectiveness of MixFilter. We attribute this to the static nature of wdecay($W_{pre}$), where the fixed penalty coefficient may be too rigid. In contrast, MixFilter's adaptive approach allows for a more nuanced adjustment, effectively balancing the preservation of pre-trained weights with the need for fine-tuning.

Finally, averaging model checkpoints along the optimization trajectory is a well-established technique to mitigate stochastic optimization noise and enhance training efficiency. This approach generally smooths out fluctuations, leading to more stable and robust performance (Polyak & Juditsky, 1992; Izmailov et al., 2018). In our experiments, we observe that applying MA complements MixFilter effectively.

**Multiplicative Gaussian Noise.** MixFilter operates by mixing pre-trained and fine-tuned weights according to Bernoulli-distributed random variables, where each variable is 1 with probability $p$ and 0 otherwise. This approach can be generalized by using other distributions, such as the Gaussian distribution, to generate the mixing variables. When applying Gaussian distribution, the method effectively perturbs the weights of the pre-trained and fine-tuned models with Gaussian noise, having a zero mean and a standard deviation equal to the weights' magnitude. This introduces variability in the weight mixing process, potentially enhancing the robustness of the model. Table 3 shows that while Gaussian sampling slightly outperforms in certain domains, on average, it is less effective compared to Bernoulli sampling.

**Comparison with activation space MixFilter.** While activation-based Dropout offers benefits such as greater flexibility in the masking strategy and the ability to apply masks on a per-sample basis rather than per mini-batch, its efficiency is limited to scenarios where activations are replaced with zeros. In contrast to weight space Dropout, leveraging pre-trained knowledge with activation-based Dropout necessitates an additional forward pass using pre-trained weights to obtain the corresponding feature maps. We set aside this computational overhead for this ablation study to directly compare the two approaches. As shown in Table 4, the advantage of activation-based MixFilter is marginal when compared to weight-based alternatives.

**Optimal placement of MixFilter.** In this experiment, we explore the optimal placement of MixFilter within the ResNet architecture. In deep neural networks, earlier layers capture more general features, while deeper layers focus on task-specific features. To determine which features are more crucial for DG, we apply

| Method | A | C | P | R | **Avg.** |
|---|---|---|---|---|---|
| Weight Space | 71.06 | 54.20 | 79.99 | 83.30 | 72.14 |
| Activation Space | 70.91 | 54.38 | 79.95 | 83.59 | 72.20 |

Table 4: Comparison of out-domain accuracy with activation and weight space MixFilter across domains from OfficeHome dataset.

| Method (ResNet18) | #Train | #Inf | A | C | P | R | **Avg.** |
|---|---|---|---|---|---|---|---|
| ERM | 1 | 1 | 54.58 | 44.62 | 69.12 | 71.43 | 59.94 |
| ERM (MA) | 1 | 1 | 58.08 | 46.16 | 70.52 | 73.72 | 62.12 |
| ENS | 18 | 18 | 57.83 | 45.96 | 71.14 | 73.84 | 62.19 |
| DiWA | 18 | 1 | 58.75 | 45.62 | 71.00 | 73.55 | 62.23 |
| **MixFilter** | 1 | 1 | 60.04 | 47.85 | 70.66 | 74.58 | 63.28 |
| **MixFilter (MA)** | 1 | 1 | 58.91 | 47.39 | 69.28 | 72.92 | 62.12 |

Table 5: Comparison of out-domain accuracy with ResNet18 backbone across domains from OfficeHome dataset.

MixFilter to convolutional layers at three different levels, referred to in Figure 3 as early layers, late layers, and across all layers.

Figure 3 shows that applying MixFilter across all convolutional layers outperforms applying it selectively. This superior performance can be attributed to the diverse characteristics of different domains. Although MixFilter on early layers generally yields better results than on late layers, certain domains, like "Real" which closely resembles the pre-trained dataset, benefit more from MixFilter applied to late layers. By employing MixFilter on all convolutional layers, we eliminate the need for nuanced, dataset-specific configurations, leading to more consistent and robust performance across varied data. It is important to note that while specific combinations of layer placements might surpass our default approach, they would introduce additional hyperparameters, complicating the tuning process—a complexity we aim to avoid.

**Different backbones.** To showcase the versatility and effectiveness of our approach across different architectures and backbones, we integrate MixFilter into ResNet18. As summarized in Table 5, the results highlight MixFilter's consistent performance gain across a range of scenarios. See Appendix B for results related to ViT.

## 5 Limitations

Although our design for MixFilter primarily drew inspiration from Dropout mechanisms in the activation space, several successful Dropout operations exist in the depth space of neural networks. Techniques like StochasticDepth (Huang et al., 2016) and DropPath (Larsson et al., 2016) have provided better results compared to Standard Dropout, particularly in ultra-deep neural networks. In this paper, we have not explored these methods as integrating them efficiently with pre-training is not trivial and requires further investigation.

Previous works have (Zoph et al., 2018; Ghiasi et al., 2018) shown that using a curriculum-based Dropout approach can enhance final performance. Specifically, starting with a low drop rate and gradually increasing it throughout training yields better results. While a fixed schedule has proven effective in our work, we believe that developing a tailored curriculum could further benefit DG tasks. Tasks that deviate significantly from the pre-training dataset might require a warm-up phase before regularization is applied. We plan to explore this in future work.

# 6    Conclusion

In DG benchmarks, ensemble-based methods consistently outperform non-ensemble approaches. This performance disparity is largely due to the richer and more robust representations that ensembles generate. They capitalize on pre-trained models and the stochastic nature of training to enhance generalization across different domains. However, these gains come with a cost: realizing the full potential of ensemble methods typically necessitates training multiple models, each with varying initializations and hyperparameters. This increased computational demand can be a significant consideration in the training of such models. In this study, we introduce MixFilter to yield the same performance of ensemble-based models without the need for multiple-model training. MixFilter makes changes to the Standard Dropout mechanism according to three core principles. First, recognizing the critical role of pre-trained knowledge in DG, MixFilter enhances Dropout by integrating this information during the masking process. Second, it adapts the masking strategy of Standard Dropout to better exploit the convolutional structure. Finally, unlike Standard Dropout which operates in the activation space, MixFilter optimizes computational efficiency by operating in the weight space. Through extensive ablation studies, we validate our design decisions and empirically demonstrate that MixFilter performs comparably with ensemble-based approaches on the DomainBed benchmark, all without introducing additional inference or training overhead.

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

# A    Implementation details

| Hyperparameter | Search Space |
|---|:---:|
| batch size | 32 |
| learning rate | {1e-5, 3e-5, 5e-5} |
| ResNet dropout | {0.0, 0.1, 0.5} |
| weight decay | {1e-4, 1e-6} |
| MixFilter mixrate | {0.6, 0.7, 0.8, 0.9} |

Table 6: Hyperparameters used for all methods in and their respective distributions for grid search.

The evaluation protocol by Gulrajani & Lopez-Paz (2020) is computationally too expensive, therefore we
use the reduced search space from Cha et al. (2021) for the common hyperparameters. Table 6 summarizes
the hyperparameter search space. We use the same search space for all datasets. To further reduce the
hyperparameter search, we find the optimal one for ERM and then use those to find the best one for each
method.

## A.1    Datasets

**PACS:** Li et al. (2017) is a 7-way object classification task with 4 domains: art, cartoon, photo, and sketch,
with $9,991$ samples.

**VLCS:** Fang et al. (2013) is a 5-way classification task from 4 domains: Caltech101, LabelMe, SUN09, and
VOC2007. There are $10,729$ samples. This dataset mostly contains real photos. The distribution shifts are
subtle and simulate real-life scenarios well.

**OfficeHome:** Venkateswara et al. (2017) is a 65-way classification task depicting everyday objects from 4
domains: art, clipart, product, and real, with a total of $15,588$ samples.

**TerraIncognita:** Beery et al. (2018) is a 10-way classification problem of animals in wildlife cameras, where
the 4 domains are different locations, L100, L38, L43, L46. There are $24,788$ samples. This represents a
realistic use case where generalization is indeed critical.

**DomainNet:** Peng et al. (2019) is a 345-way object classification task from 6 domains: clipart, infograph,
painting, quickdraw, real, and sketch. With a total of $586,575$ samples, it is larger than most of the other
evaluated datasets in both samples and classes.

## B    MixFilter For ViT

While we designed MixFilter with ConvNets as the backbone, the core principles outlined in section 3
are broadly applicable and can be adapted to various architectures. To demonstrate the versatility of our

| Method (ViT-S/16) | #Train | #Inf | A | C | P | R | **Avg.** |
|---|---|---|---|---|---|---|---|
| ERM | 1 | 1 | 69.26 | 51.60 | 78.07 | 81.87 | 70.20 |
| ERM (MA) | 1 | 1 | 72.09 | 55.30 | 80.83 | 83.19 | 72.85 |
| ENS | 6 | 6 | 71.37 | 55.78 | 79.95 | 83.22 | 72.58 |
| DiWA | 6 | 1 | 72.25 | 55.78 | 80.12 | 82.99 | 72.78 |
| **MixFilter** | 1 | 1 | 71.78 | 54.32 | 80.55 | 82.19 | 72.21 |
| **MixFilter (MA)** | 1 | 1 | 72.55 | 54.98 | 80.41 | 82.53 | 72.62 |

Table 7: Comparison of out-domain accuracy with (ViT-S/16) Dosovitskiy et al. (2020) backbone across domains from OfficeHome dataset.

approach, we extended MixFilter to the ViT Dosovitskiy et al. (2020) architecture. Compared to ResNet, ViT has a different structure, and in line with the second principle, an effective Dropout mechanism for DG tasks should regulate the flow of information throughout the network. To accomplish this, we adopted the same masking mechanism as Mixout, which is better suited to the ViT architecture. We applied Mixout to all linear layers within the ViT, including both the MLP and attention layers, while retaining the rest of MixFilter's original design. As shown in Table 7, MixFilter consistently outperforms the baselines, highlighting the robustness and generality of our design principles for DG.

## C    Full Results

In this section, we show detailed results of Table 1 of the main manuscript. Tables 8, 9, 10, 11 12 show full results on PACS, VLCS, OfficeHome, TerraIncognita, and DomainNet datasets, respectively. The provided tables summarize the obtained out-of-distribution accuracy for every domain within the five datasets. Standard deviations are reported with different seeds when possible. To guarantee the comparability of the results, we followed the same experimental setting as in DomainBed (Gulrajani & Lopez-Paz, 2020).

| Method | #Train | #Inf | A | C | P | S | **Avg.** |
|---|---|---|---|---|---|---|---|
| Large Dropout | *1* | *1* | $87.78_{\pm1.31}$ | $82.68_{\pm0.28}$ | $98.43_{\pm0.15}$ | $79.66_{\pm0.75}$ | $87.14_{\pm0.62}$ |
| ERM | *1* | *1* | $90.81_{\pm0.87}$ | $81.68_{\pm0.78}$ | $98.68_{\pm0.26}$ | $79.45_{\pm0.94}$ | $87.66_{\pm0.71}$ |
| CORAL | *1* | *1* | $89.36_{\pm0.76}$ | $80.44_{\pm0.99}$ | $98.58_{\pm0.11}$ | $81.23_{\pm0.82}$ | $87.40_{\pm0.67}$ |
| ERM (MA) | *1* | *1* | $91.68_{\pm0.21}$ | $82.84_{\pm0.20}$ | $98.90_{\pm0.07}$ | $79.57_{\pm1.04}$ | $88.25_{\pm0.38}$ |
| ENS | *18* | *18* | 90.85 | 83.53 | 98.88 | 82.95 | 89.05 |
| DiWA | *18* | *1* | 92.01 | 84.01 | 99.18 | 81.65 | 89.21 |
| **MixFilter** | *1* | *1* | $89.57_{\pm1.23}$ | $83.96_{\pm0.91}$ | $98.85_{\pm0.07}$ | $79.38_{\pm1.26}$ | $87.94_{\pm0.87}$ |
| **MixFilter (MA)** | *1* | *1* | $91.07_{\pm0.14}$ | $83.46_{\pm0.18}$ | $99.15_{\pm0.04}$ | $79.32_{\pm0.50}$ | $88.25_{\pm0.22}$ |

Table 8: Out-of-domain accuracies (%) on PACS.

| Method | #Train | #Inf | C | L | S | V | **Avg.** |
|---|---|---|---|---|---|---|---|
| Large Dropout | *1* | *1* | $97.76_{\pm0.46}$ | $64.82_{\pm0.35}$ | $74.41_{\pm0.38}$ | $80.25_{\pm0.54}$ | $79.31_{\pm0.43}$ |
| ERM | *1* | *1* | $98.06_{\pm0.15}$ | $64.28_{\pm0.49}$ | $76.72_{\pm0.48}$ | $79.48_{\pm0.60}$ | $79.64_{\pm0.43}$ |
| CORAL | *1* | *1* | $98.82_{\pm0.10}$ | $64.94_{\pm0.69}$ | $76.83_{\pm0.77}$ | $79.46_{\pm0.52}$ | $80.01_{\pm0.52}$ |
| DiWA | *18* | *1* | 98.06 | 63.67 | 76.96 | 89.64 | 79.83 |
| ERM (MA) | *1* | *1* | $98.09_{\pm0.13}$ | $64.11_{\pm0.35}$ | $77.58_{\pm0.28}$ | $79.66_{\pm0.52}$ | $79.86_{\pm0.32}$ |
| ENS | *18* | *18* | 98.06 | 64.89 | 76.28 | 80.90 | 80.03 |
| **MixFilter** | *1* | *1* | $98.20_{\pm0.10}$ | $65.68_{\pm0.12}$ | $73.88_{\pm0.55}$ | $79.85_{\pm0.80}$ | $79.40_{\pm0.39}$ |
| **MixFilter (MA)** | *1* | *1* | $98.50_{\pm0.07}$ | $62.85_{\pm0.15}$ | $74.74_{\pm0.16}$ | $81.93_{\pm0.54}$ | $79.50_{\pm0.23}$ |

Table 9: Out-of-domain accuracies (%) on VLCS.

| Method | #Train | #Inf | A | C | P | R | **Avg.** |
|---|---|---|---|---|---|---|---|
| ERM | *1* | *1* | $68.95_{\pm1.16}$ | $52.13_{\pm0.67}$ | $78.61_{\pm0.48}$ | $82.14_{\pm0.49}$ | $70.46_{\pm0.70}$ |
| CORAL | *1* | *1* | $70.08_{\pm0.50}$ | $53.20_{\pm0.39}$ | $78.95_{\pm0.29}$ | $82.69_{\pm0.18}$ | $71.23_{\pm0.34}$ |
| Large Dropout | *1* | *1* | $68.64_{\pm0.74}$ | $53.30_{\pm0.30}$ | $78.13_{\pm0.43}$ | $82.56_{\pm0.05}$ | $70.66_{\pm0.38}$ |
| DiWA | *18* | *1* | 70.55 | 53.64 | 79.76 | 83.02 | 71.74 |
| ENS | *18* | *18* | 69.77 | 54.04 | 79.95 | 83.33 | 71.77 |
| ERM (MA) | *1* | *1* | $71.27_{\pm0.24}$ | $53.67_{\pm0.02}$ | $79.50_{\pm0.30}$ | $83.43_{\pm0.10}$ | $71.97_{\pm0.16}$ |
| **MixFilter** | *1* | *1* | $71.06_{\pm0.68}$ | $54.20_{\pm0.35}$ | $79.99_{\pm0.11}$ | $83.30_{\pm0.06}$ | $72.14_{\pm0.30}$ |
| **MixFilter (MA)** | *1* | *1* | $72.47_{\pm0.24}$ | $54.81_{\pm0.29}$ | $79.74_{\pm0.24}$ | $83.62_{\pm0.17}$ | $72.66_{\pm0.24}$ |

Table 10: Out-of-domain accuracies (%) on OfficeHome.

| Method | #Train | #Inf | L100 | L38 | L43 | L46 | **Avg.** |
|---|---|---|---|---|---|---|---|
| ERM | *1* | *1* | $59.53_{\pm2.79}$ | $48.93_{\pm1.79}$ | $61.87_{\pm1.57}$ | $40.13_{\pm3.17}$ | $52.62_{\pm2.33}$ |
| CORAL | *1* | *1* | $58.21_{\pm1.94}$ | $47.59_{\pm1.62}$ | $57.65_{\pm0.51}$ | $38.98_{\pm1.84}$ | $50.61_{\pm1.48}$ |
| Large Dropout | *1* | *1* | $61.31_{\pm2.79}$ | $47.65_{\pm1.99}$ | $60.71_{\pm0.64}$ | $39.41_{\pm0.77}$ | $52.27_{\pm1.55}$ |
| ENS | *18* | *18* | 63.67 | 46.44 | 63.48 | 42.83 | 54.10 |
| ERM (MA) | *1* | *1* | $61.46_{\pm1.53}$ | $50.10_{\pm0.96}$ | $63.58_{\pm0.29}$ | $43.23_{\pm0.63}$ | $54.59_{\pm0.85}$ |
| DiWA | *18* | *1* | 62.98 | 50.44 | 62.47 | 46.85 | 55.68 |
| **MixFilter** | *1* | *1* | $65.53_{\pm0.49}$ | $56.93_{\pm1.32}$ | $64.27_{\pm0.21}$ | $46.95_{\pm0.64}$ | $58.42_{\pm0.66}$ |
| **MixFilter (MA)** | *1* | *1* | $63.82_{\pm0.50}$ | $58.84_{\pm0.81}$ | $63.70_{\pm0.20}$ | $47.61_{\pm0.73}$ | $58.49_{\pm0.56}$ |

Table 11: Out-of-domain accuracies (%) on TerraIncognita.

| Method | #Train | #Inf | clip | info | paint | quick | real | sketch | **Avg.** |
|---|---|---|---|---|---|---|---|---|---|
| ERM | 1 | 1 | $67.09_{\pm 0.10}$ | $25.58_{\pm 0.32}$ | $56.21_{\pm 0.97}$ | $14.85_{\pm 0.32}$ | $69.56_{\pm 0.52}$ | $57.61_{\pm 0.62}$ | $48.48_{\pm 0.48}$ |
| CORAL | 1 | 1 | $66.89_{\pm 0.20}$ | $24.43_{\pm 0.20}$ | $54.50_{\pm 0.26}$ | $13.82_{\pm 0.27}$ | $68.34_{\pm 0.31}$ | $56.02_{\pm 0.48}$ | $47.33_{\pm 0.29}$ |
| Large Dropout | 1 | 1 | $67.04_{\pm 0.10}$ | $25.14_{\pm 0.28}$ | $54.48_{\pm 0.11}$ | $13.37_{\pm 0.26}$ | $68.22_{\pm 0.15}$ | $55.90_{\pm 0.12}$ | $47.36_{\pm 0.17}$ |
| DiWA | 18 | 1 | 66.69 | 25.15 | 56.73 | 14.66 | 70.40 | 56.79 | 48.40 |
| ERM (MA) | 1 | 1 | $66.99_{\pm 0.02}$ | $26.03_{\pm 0.05}$ | $57.57_{\pm 0.14}$ | $15.16_{\pm 0.07}$ | $70.19_{\pm 0.06}$ | $58.34_{\pm 0.04}$ | $49.05_{\pm 0.06}$ |
| ENS | 18 | 18 | 68.66 | 25.39 | 56.99 | 14.58 | 71.28 | 57.74 | 49.11 |
| **MixFilter** | 1 | 1 | $66.78_{\pm 0.30}$ | $24.26_{\pm 0.21}$ | $54.90_{\pm 0.26}$ | $14.24_{\pm 0.28}$ | $69.13_{\pm 0.18}$ | $56.84_{\pm 0.36}$ | $47.69_{\pm 0.26}$ |
| **MixFilter (MA)** | 1 | 1 | $66.44_{\pm 0.05}$ | $25.86_{\pm 0.06}$ | $57.51_{\pm 0.08}$ | $15.15_{\pm 0.08}$ | $70.04_{\pm 0.02}$ | $58.36_{\pm 0.03}$ | $48.89_{\pm 0.05}$ |

Table 12: Out-of-domain accuracies (%) on DomainNet.

