# OpenReview forum: "MixFilter: Pre-train Aware Structured Dropout for Domain Generalization"
_TMLR — Rejected by TMLR_

### Review · Reviewer_uXoU · 2024-08-29

**Summary Of Contributions:**

This paper introduces the MixFilter method for domain generalization. MixFilter employs several dropout-like techniques to blend pre-trained and fine-tuned models, enabling the ensemble of multiple sub-networks while operating within a single model. This approach reduces computational costs and enhances model robustness when comparing to traditional ensembling methods.

**Audience:**

Yes

**Claims And Evidence:**

Yes

**Requested Changes:**

1. Additional ablation studies on several components are necessary.
2. How might the concepts behind MixFilter apply to other network backbones?
3. Further investigation may be needed to fully understand MixFilter.

**Strengths And Weaknesses:**

The proposed MixFilter method is simple and shows some improvements over the baselines, which is positive. However, it lacks a deep foundational understanding. Additionally, the method is only tested on ConvNets, which may limit its applicability as a general solution for addressing the domain generalization problem.

---

> ### Author Response · Authors · 2024-10-20
> **Response to Reviewer uXoU**
>
> > How might the concepts behind MixFilter apply to other network backbones?
>
> We would like to clarify that our initial aim was to adhere closely to the DomainBed benchmark by evaluating MixFilter on the ResNet50 backbone. However, to highlight the versatility and effectiveness of our approach across different ConvNet architectures, we have now expanded our ablation studies to include ResNet18. Furthermore, although MixFilter was initially designed with ConvNets as its backbone, the core principles outlined in section 3 are broadly applicable and adaptable to a variety of architectures. To illustrate this flexibility, we also extended MixFilter to the ViT architecture. Across all scenarios, we observed consistent performance improvements in different settings.
>
> > Additional ablation studies on several components are necessary / Further investigation may be needed to fully understand MixFilter.
>
> To better understand MixFilter, we conducted extensive ablation studies examining various aspects of the proposed method. This analysis included comparisons with existing Dropout variants, special fine-tuning techniques, the impact of noise, the placement of MixFilter on different layers, and the effect of MixFilter in the activation space versus the weight space. Across all these cases, we consistently observed that MixFilter either complements existing methods or shows clear improvements over them. While we acknowledge that our understanding of MixFilter is still evolving, we kindly ask the reviewer to clarify their question so that we can address it more effectively.

---

### Review · Reviewer_9gcY · 2024-10-04

**Summary Of Contributions:**

The paper introduces a novel method for domain generalization (DG) in CNN models that obtains similar performance to state-of-the-art ensemble training methods that do not require training multiple models. The authors adapt existing methods for dropout in CNNs, in particular structured masking, and then leverage both the currently finetuned model + pretrained model in weight space to prevent multiple forward passes of two different models. Extensive experimental results show that MixFilter consistently performs similarly compared to state-of-the-art ensemble DG methods.

**Audience:**

Yes

**Claims And Evidence:**

Yes

**Requested Changes:**

- [critical] it is presented in the abstract and intro that ensemble methods outperform non-emsemble methods, however table 1 appears to show very little difference in OOD performance whether ensemble was used or not. Can this be addressed?
- [critical] a key motivation for the MixFilter method is that it is computation efficient compared to ensemble methods. However, the evaluation is missing measurements on the difference in efficiency between existing methods and the one being proposed. Can the authors include benchmarks results on the speed of MixFilter vs existing methods (both ensemble and not-ensemble)?
- [non-critical] Mixout was included in figure 2, but not included in table 1, can you include Mixout results in table 1?

**Strengths And Weaknesses:**

**Strengths**

- strong and convincing evaluation setup of MixFilter OOD performance compared to other methods
- using weight space instead of activation space to prevent need for two forward passes is clever
- intuitive design that combines multiple existing techniques for domain generalization

**Weaknesses**

- MixFilter efficiency improvement over ensemble methods is not measured
- only one pretrained model is tested

---

> ### Author Response · Authors · 2024-10-20
> **Response to Reviewer 9gcY**
>
> > table 1 appears to show very little difference in OOD performance whether ensemble was used or not. Can this be addressed?
>
> We emphasize that ensemble-based methods can be categorized into two distinct families. Explicit ensembles, like ENS and DiWA, construct the ensemble by training each model separately. In contrast, implicit ensembles, such as ERM (MA), generate the ensemble within a single training run as an exponential moving average of the model weights. As discussed in the main manuscript and supported by [1], MixFilter belongs to the latter category. Based on this classification, we observe that, on average, all ensemble-based methods outperform their non-ensemble counterparts (such as ERM and CORAL).
>
> > Can the authors include benchmarks results on the speed of MixFilter vs existing methods
>
> Thank you for your feedback. In the original manuscript, we presented the computational aspects of different methods in Table 1. Specifically, the columns labeled #Train and #Inf indicate the number of models required by each method during training and inference. Notably, unlike ENS and DiWA which use 18 models, MixFilter requires only a single model during training, thus the training time can be highly reduced. We have expanded Section 4.1 to further clarify this point.
>
> > Can you include Mixout results in table 1?
>
> In response to the reviewers' request, we have added the Mixout results to Table 1.
>
> > Only one pretrained model is tested
>
> We would like to clarify that our initial aim was to adhere closely to the DomainBed benchmark by evaluating MixFilter on the ResNet50 backbone. However, to highlight the versatility and effectiveness of our approach across different ConvNet architectures, we have now expanded our ablation studies to include ResNet18. Furthermore, although MixFilter was initially designed with ConvNets as its backbone, the core principles outlined in section 3 are broadly applicable and adaptable to a variety of architectures. To illustrate this flexibility, we also extended MixFilter to the ViT architecture. Across all scenarios, we observed consistent performance improvements in different settings
>
> *[1] Srivastava, Nitish, et al. "Dropout: a simple way to prevent neural networks from overfitting." The journal of machine learning research 15.1 (2014): 1929-1958.*

---

> > ### Comment · Reviewer_9gcY · 2024-11-12
> > **Thank you for rebuttal**
> >
> > I would like to thank the authors for their responses. In general, I think most of my comments have been addressed.

---

### Review · Reviewer_S26H · 2024-10-15

**Summary Of Contributions:**

The authors propose a structured dropout method called MixFilter by dropping out whole channels. Their method performs slightly better than competing approaches on the DomainBed benchmark.

**Audience:**

Yes

**Claims And Evidence:**

No

**Requested Changes:**

Address issues listed in weaknesses above.

**Strengths And Weaknesses:**

strengths:
- The paper is clearly written and the experimental results show some improvements on the DomainBed benchmark.

- The authors carry out detailed ablation studies on various dropout variants.


weaknesses:
- The biggest issue of the paper is the lack of novelty. It is essentially SpatialDropout applied to a domain generalization dataset. I struggle to find new ideas in the algorithm.

- The empirical improvements on the DomainBed benchmark in Table 1 is also quite small, compared to say ERM or ERM (MA). Most of the improvements come from one single dataset TerraInc. Can the authors explain why is there such a big improvement for that dataset?

---

> ### Author Response · Authors · 2024-10-20
> **Response to Reviewer S26H**
>
> > It is essentially SpatialDropout applied to a domain generalization dataset. I struggle to find new ideas in the algorithm.
>
> In our ablation studies, we systematically evaluated various Dropout variants on DG benchmarks, and none of them were found to address the core principles outlined in section 3 individually. Notably, applying SpatialDropout to all layers of the backbone network resulted in an even poorer performance than the ERM baseline. This decline occurs because SpatialDropout zeroes out entire neurons during each forward pass, introducing complex noise that disrupts the backbone network's training. These observations led us to develop MixFilter, specifically designed to overcome the limitations of existing Dropout mechanisms in the context of DG tasks.
>
> > Most of the improvements come from one single dataset TerraInc. Can the authors explain why is there such a big improvement for that dataset?
>
> Building on the findings of [1], we know that TerraInc exhibits the highest feature diversity shift among the datasets in the DomainBed benchmark, indicating significant variation in input features across its domains. Furthermore, as highlighted by [2], richer representations with redundant information are particularly effective in scenarios with high feature diversity shifts. MixFilter leverages this insight by stochastically blending pre-trained knowledge with downstream knowledge to induce more robust representations. Additionally, the greater performance gains of ensemble-based methods (such as ERM (MA), ENS, and DiWA) on the TerraInc dataset further validate our hypothesis.
>
>
> *[1] Chen, Yimeng, et al. "Explore and exploit the diverse knowledge in model zoo for domain generalization." International Conference on Machine Learning. PMLR, 2023.*
>
> *[2] Zhang, Jianyu, and Léon Bottou. "Learning useful representations for shifting tasks and distributions." International Conference on Machine Learning. PMLR, 2023.*

---

### Author Response · Authors · 2024-10-20
**Global Response**

We sincerely thank the reviewers for their detailed and valuable feedback on our manuscript. We're pleased that they found our paper to be clear and well-structured, with a thorough evaluation setup and ablation studies, and that they recognized the MixFilter design as intuitive and effective in integrating multiple techniques for domain generalization.  We have made revisions and additions to address your concerns in response to your suggestions. Below, we outline the key improvements and new results that we believe will be of particular interest:
1. By identifying key principles in section 3 for a successful DG method, we developed MixFilter as a Dropout variant that adheres to these design principles. To ensure alignment with the DomainBed benchmark, we conducted a thorough evaluation using the ResNet50 backbone. Based on the reviewers' request, we have now expanded our ablation studies to include ResNet18. Although MixFilter was initially designed for ConvNet backbones, the core principles described broadly apply to a range of architectures. To demonstrate this versatility, we also extended MixFilter to the ViT architecture. In all scenarios, we observed consistent performance improvements, underscoring the adaptability and effectiveness of our approach.
2. We have expanded Section 4.1 to further clarify the computational advantages of MixFilter compared to ensemble-based baselines.

---

### Decision · Action_Editor_qt2y · 2024-11-25

**Recommendation:** Reject

**Comment:**

The paper introduces an ensembling strategy for convolutional neural networks that leverages dropout across channels. Through the use of structured masking, the approach ensembles a pre-trained and fine-tuned network within a single forward pass, avoiding the need for multiple passes.
While reviewers acknowledged the method's clarity and simplicity, they questioned whether the paper provides sufficient insights. On the evaluated benchmarks, simple baseline methods, such as ERM, demonstrate comparable performance to the proposed algorithm while maintaining similar computational efficiency.

**Audience:**

Yes, out-of-domain generalization of vision models is certainly a topic of interest for the TMLR audience.

**Claims And Evidence:**

The paper claim to achieve performance "comparable to ensemble-based approaches while avoiding additional inference or training overhead." However, reviewers pointed out that baseline methods (ERM) achieve comparable performance using the same amount of training and inference models. While the authors' claim is technically not wrong, I think it is overstated.

**Resubmission Of Major Revision:**

The authors may consider submitting a major revision at a later time.